# Using a learning health system framework to examine COVID-19 pandemic planning and response at a Canadian Health Centre

Christine Cassidy[1,2], Meaghan Sim[3], Mari Somerville [1], Daniel Crowther[1], Douglas Sinclair[2], Annette Elliott Rose[2], Stacy Burgess[2], Shauna Best[2], Janet A. Curran[1,2] *

1 School of Nursing, Faculty of Health, Dalhousie University, Halifax, NS, Canada, 2 Izaak Walton Killam (IWK) Health Centre, Halifax, NS, Canada, 3 Research, Innovation & Discovery, Nova Scotia Health, Halifax, NS, Canada

* jacurran@dal.ca

## Abstract

### Background

The COVID-19 pandemic has presented a unique opportunity to explore how health systems adapt under rapid and constant change and develop a better understanding of health system transformation. Learning health systems (LHS) have been proposed as an ideal structure to inform a data-driven response to a public health emergency like COVID-19. The aim of this study was to use a LHS framework to identify assets and gaps in health system pandemic planning and response during the initial stages of the COVID-19 pandemic at a single Canadian Health Centre.

### Methods

This paper reports the data triangulation stage of a concurrent triangulation mixed methods study which aims to map study findings onto the LHS framework. We used a triangulation matrix to map quantitative (textual and administrative sources) and qualitative (semi-structured interviews) data onto the seven characteristics of a LHS and identify assets and gaps related to health-system receptors and research-system supports.

### Results

We identified several health system assets within the LHS characteristics, including appropriate decision supports and aligned governance. Gaps were identified in the LHS characteristics of engaged patients and timely production and use of research evidence.

### Conclusion

The LHS provided a useful framework to examine COVID-19 pandemic response measures. We highlighted opportunities to strengthen the LHS infrastructure for rapid integration of evidence and patient experience data into future practice and policy changes.

**Data Availability Statement:** All relevant data are within the paper and its Supporting Information files.

**Funding:** This research (author JC) received financial support from the Nova Scotia COVID-19 Health Research Coalition. (URL: https://researchns.ca/covid19-health-research-coalition/) The funders had no role in study design, data collection and analysis, decision to publish, or preparation of the manuscript.

**Competing interests:** The authors have declared that no competing interests exist.

## Introduction

The COVID-19 pandemic has been described as an extreme stress test of the health system and society at large, resulting in widespread organizational and societal changes [1]. In March 2020, health systems implemented a range of policies and protocols to reduce the risk of disease transmission in hospital and clinical settings, including patient visitor restrictions, physical distancing measures, and enhanced personal protective equipment [2]. Further, health system interventions were rapidly designed and implemented to address gaps in care caused by the pandemic measures. For example, telemedicine and virtual care options were quickly rolled out across a number of specialties for non-urgent care [3, 4]. These efforts brought rapid change to a health system that has been known to be slow to transform [5, 6].

COVID-19 has presented an unparalleled opportunity to explore how health systems adapt under rapid and constant change. Such insights are valuable for informing the development of resilient and sustainable systems. Learning health systems (LHS) have been proposed as an ideal structure to inform a data-driven response to a public-health emergency like COVID-19 [7]. A LHS is an environment in which *"science, informatics, incentives and culture are aligned for continuous improvement and innovation, with best practices seamlessly embedded in the delivery process and new knowledge captured as an integral by-product of the delivery experience"* [8]. Learning cycles are the fundamental processes of LHS which seek to strike a balance between patient and provider experiences and health system costs [9]. A LHS is able to respond rapidly to changing evidence and incorporate lessons learned from patient experiences on a continuous basis. There is deliberate overlap between clinical practice, quality improvement, and research and innovation [10]. This structure is critical to accelerate the most up-to-date research into real-world practice.

LHS have shown to catalyze an efficient and effective health system, [11] including: improved patient outcomes and experiences; better healthcare provider training and experience; optimized use of evidence for health system decision-making; and more equitable healthcare [12]. Despite the value, few health service organizations have actualized a LHS in practice [13]. The literature describes theoretical conceptualizations of LHS but lack description of how to best design and implement the components of a LHS [14–16]. To actualize the benefits of a LHS and transform healthcare systems, there is a need for practical guidance on how to best use a LHS in practice.

COVID-19 has changed the course of health care and has been identified as an excellent case for highlighting the urgent need to develop LHS [17, 18]. Given the rapidly evolving response required for COVID-19, a LHS framework can offer a structure for examining continuous learning and improvement during pandemic planning and response. Further, implementation research has a crucial role to play in identifying important barriers and enablers to the development of a LHS and tailoring interventions to support its use in practice [18]. As such, the overall goal of this study was to use a LHS framework [9] to examine the pandemic planning and preparedness work operationalized at a Canadian women and children's tertiary health centre. This paper aims to examine the utility of a LHS framework for examining rapid learning and change during the COVID-19 pandemic. Specific objectives include:

i. Map pandemic planning and response resources and strategies onto a LHS framework

ii. Identify assets and gaps in the COVID-19 pandemic planning and preparedness work

iii. Describe how a LHS can be used as a framework to inform health system change.

## Methods

### Study design

This paper reports the data triangulation stage of a mixed methods research study. For the larger study, we used a concurrent triangulation mixed methods design [19] to integrate qualitative data and quantitative data from various sources through iterative cycles of data collection, data confirmation, and data analysis. Findings from the quantitative and qualitative strands of this study are reported elsewhere, while this paper reports how the study findings were mapped onto the LHS framework. Integration of different data sources allows for a clearer understanding of the research phenomenon than either quantitative or qualitative research alone [19]. In this study, the purpose of the quantitative data was to help contextualize the qualitative interview findings.

Guided by an integrated knowledge translation approach, [20] our team of clinicians, managers, researchers and administrators met fortnightly throughout the project to discuss project milestones, preliminary impressions, gaps in data collection and data analysis. Project summaries were also distributed to all team members via email every two weeks. Research ethics approval (Institutional approval #1025812) was obtained prior to commencement of data collection.

### Framework

This study used a LHS framework to map the assets and gaps in the COVID-19 pandemic planning and preparedness work at a women's and children's tertiary hospital. Lavis et al. defined a rapid-learning health system as a combination of a health system and a research system that is: 1) anchored in patient needs, perspectives and aspirations; 2) driven by timely data and evidence; 3) supported by appropriate decision supports and aligned governance, financial and delivery arrangements; and 4) enabled with a culture of rapid learning and improvement [9]. Lavis and colleagues expand on this definition by defining seven LHS characteristics (Table 1) and have used these characteristics to map assets and gaps for creating rapid-learning health systems in 14 Canadian jurisdictions [9]. We employed a similar mapping approach to examine assets and gaps in COVID-19 pandemic planning and preparedness work at one health centre.

### Study setting

This study was conducted at a single, Canadian tertiary health centre serving children, youth, and women from the four Atlantic Canadian provinces (hereinafter referred to as the 'Health Centre'). It is a publicly funded health system serving rural and urban populations across the four Atlantic Canadian provinces. Atlantic Canada has a poorer health profile and higher rates of poverty than the rest of Canada [21]. In 2020/21, the Health Centre reported 22,690 emergency department visits, 256,320 ambulatory clinic visits, 12,970 acute inpatient admissions, and 61,163 virtual appointments [22]. The World Health Organization defines a health system as "the institutions, people, and resources involved in delivering health care to individuals" [23, p. 105]. This may include large-scale systems, such as the Canadian health care system, or smaller institutions, such as a single hospital or health authority. For the purpose of this study, the Health Centre is the defined health system.

### Data triangulation

**Triangulation data sources.** We used a LHS framework to triangulate data from the quantitative and qualitative strands of the larger mixed methods study (Fig 1; Table 2). The quantitative strand data includes administrative and textual sources such as relevant organizational documents and system performance data generated between January 1st, 2020, and

**Table 1. Learning health system characteristics [9].**

| LHS Characteristic | Definition |
|---|---|
| Engaged patients | Systems are anchored on patient needs, perspectives and aspirations (at all levels) and focused on improving their care experiences and health at manageable per capita costs and with positive provider experiences. |
| Digital capture, linkage and timely sharing of relevant data | Systems capture, link and share (with individuals at all levels) data (from real-life, not ideal conditions) about patient experiences (with services, transitions and longitudinally) and provider engagement alongside data about other process indicators (e.g., clinical encounters and costs) and outcome indicators (e.g., health status). |
| Timely production of research evidence | Systems produce, synthesize, curate and share (with individuals at all levels) research about problems, improvement options and implementation considerations. |
| Appropriate decision supports | Systems support informed decision-making at all levels with appropriate data, evidence, and decision-making frameworks. |
| Aligned governance, financial and delivery arrangements | Systems adjust who can make what decisions (e.g., about joint learning priorities), how money flows and how the systems are organized and aligned to support rapid learning and improvement at all levels. |
| Culture of rapid learning and improvement | Systems are stewarded at all levels by leaders committed to a culture of teamwork, collaboration and adaptability. |
| Competencies for rapid learning and improvement | Systems are rapidly improved by teams at all levels who have the competencies needed to identify and characterize problems, design data- and evidence-informed approaches (and learn from other comparable programs, organizations, regions, and sub-regional communities about proven approaches), implement these approaches, monitor their implementation, evaluate their impact, make further adjustments as needed, sustain proven approaches locally, and support their spread widely. |

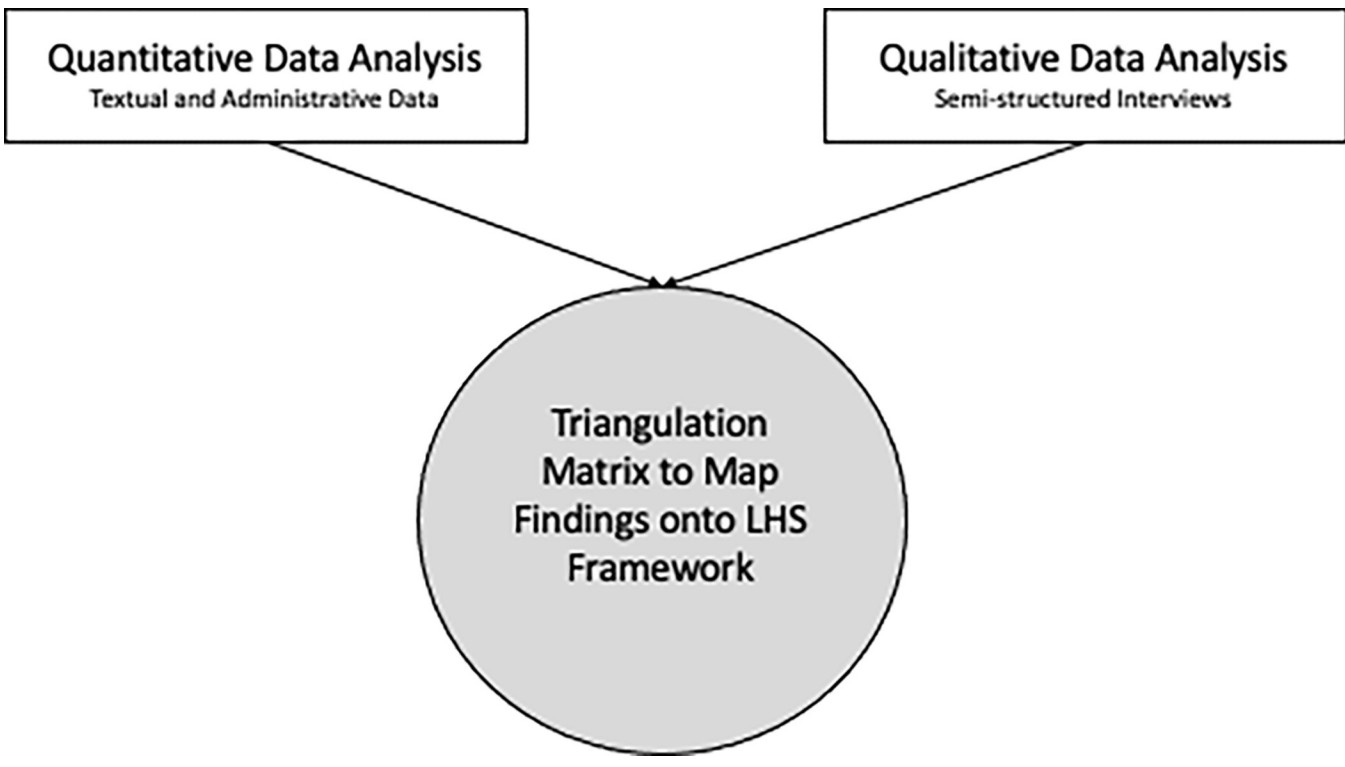

**Fig 1. Mixed methods study flow diagram.**

**Table 2. Data collection for administrative, textual, and qualitative interview data [24].**

| Administrative Data | Textual Data | Qualitative Interview Data |
|---|---|---|
| *Health administrative Data*:<br>• monthly admissions and discharges from the different inpatient and emergency care areas<br>• visits/cancellations at ambulatory clinics<br>• surgical scheduling (cancellations and scheduled surgeries),<br>• diagnostic imaging | Health Centre communication mechanisms including town halls, newsletters, intranet COVID-19 Subsite, email announcements, social media (i.e., Facebook, Twitter, Instagram) | *Patient and Caregiver Participants*:<br>• Twenty-one patient and caregiver interviews were conducted between July 9th and August 12th, 2020.<br>• Ten participants were parents of children who received care at the Health Centre during the pandemic, four were adult patients and seven were women who experienced labour and delivery during the study time frame. |
| *Human Resource Data*:<br>• staff attendance<br>• hiring<br>• leave of absences<br>• special paid leave<br>• termination<br>• redeployment | New and revised institutional clinical care or operational decisions, directives and policies related to COVID-19 | *Health Centre Staff Participants*:<br>• Thirty-three staff interviews were conducted between July 20th and August 24th, 2020.<br>• Six of these participants held leadership positions at the Health Centre.<br>• Other participant roles included: Administrative assistants, child life specialist, clinical coordinator, director, door screener, food service supervisor, genetic counsellor, manager, medical lab technologist, nurse practitioner, occupational therapist, payroll time administrator, physician, recreational therapist, registered nurse, speech language pathologist. |
| *Technology/Operational Support and Supplies Data*:<br>• newly developed PPE project models,<br>• information technology device ordering and loans<br>• incidents and service requests at information technology support; | New and revised Health Centre department specific pandemic response documents | |
| | Meeting notes of special committees that were convened in response to the pandemic meeting notes of pre-existing committees that discussed COVID-19 response ( | |
| | Health Centre COVID Dashboard | |
| | Provincial Health Protections Act Order | |

August 31st, 2020. This included health administrative and human resource data; policies and directives developed or adapted in response to the pandemic; health centre communications; town hall meeting notes; and meeting notes from special committees convened in response to COVID-19. The qualitative data includes six themes related to key pandemic response priorities that were generated from semi-structured interviews with patients and families, health care providers, leadership and management team, and operations and support workers. The six themes are: 1) Access to Health Centre, 2) Personal protective equipment (PPE), 3) Visitor Restrictions, 4) Pandemic Assessment Centre (PAC), 5) Working from Home, and 6) Food Services (Table 3) [24].

**Table 3. Six key priority areas identified in qualitative strand with corresponding definition [24].**

| Key Priority Identified | Description of Key Priority |
|---|---|
| 1. Access to health care | Encompasses any relevant data related to access to the health care which arose because of the pandemic response. This includes cancellations and closures, restrictions to labs and diagnostic imaging, the creation of the Pandemic Response Unit (PRU), and virtual care. |
| 2. Personal protective equipment (PPE) | Encompasses any relevant data related to PPE which arose during the pandemic response. This includes directives pertaining to the usage of masks and scrubs, the sourcing and storing of PPE, and the PPE-related educational efforts targeted at the staff. |
| 3. Visitor restrictions | Encompasses any relevant data related to visitor restrictions due to the pandemic response. |
| 4. Pandemic Assessment Centre (PAC) | Encompasses any relevant data pertaining to the creation, operation and changes of the PAC. |
| 5. Working from home | Encompasses any relevant data related to the transition and process of working from home. As well, it includes the IT infrastructure and changes which took place to ease the transition and process of working from home. |
| 6. Food services | Refers to any relevant data related to the closure and cancellation of Food Services and any additional food supports that were developed during the initial pandemic response. |

**Triangulation protocol.** We employed a data triangulation protocol to map the quantitative and qualitative findings onto the LHS characteristics. A triangulation protocol is a detailed approach to examine meta-themes across findings from different data components that have already been analyzed individually [25] (Farmer, Robinson, Elliott, & Eyles, 2006). First, we used the LHS framework to create a convergence-coding matrix that displayed the seven LHS characteristics in rows and two columns for the health-system receptors and research-system supports. Second, two independent members categorized the quantitative and qualitative findings into the corresponding cells. The matrix was reviewed by three additional team members and discussion was used to achieve consensus on categorizations. Initial findings were shared with the full research team during a virtual meeting to identify areas of convergence, divergence, and discrepancies among the data. Lastly, following the team discussion on verification and clarification, the matrix was finalized to reveal assets and gaps in the initial COVID-19 pandemic response as it relates to the LHS characteristics.

## Triangulation results

The following results represent the findings from the quantitative and qualitative study strands mapped onto the seven LHS characteristics and identification of assets and gaps in the COVID-19 pandemic planning and preparedness work (S1 File).

### LHS characteristic 1: Engaged patients

During the Health Centre's COVID-19 response, patients were passively engaged through the dissemination of rapidly changing information to patients and families through various channels. Social media platforms (Facebook, Twitter, and Instagram) and the Health Centre's website were the main avenues of communication with public regarding cancellations, closures, reopening of services and visitor restrictions. Despite these efforts, study findings highlight a shift from patient-centered care during the first wave of the pandemic response. For example, critical policies related to strict visitor restrictions and access to the health centre were developed and implemented by the leadership team as part of the rapid response to managing the impact of the pandemic; however, patient and family partners were not involved in this in this process. The Health Centre did launch the COVID-19 Patient Survey in August 2020 to gather feedback from patients and families about their experience throughout early stages of the pandemic response.

### LHS characteristic 2: Digital capture, linkage and timely sharing of relevant data

From the outset of the pandemic, teams worked quickly to *capture, link, and share* relevant COVID-19 data. The Health Centre developed a new structure to collect administrative data related to PAC, including volumes of patients and number of registrations. The Health Centre's Incident Management Committee (IMC) tracked and used PAC administrative data to inform decisions regarding redeployment to PAC, required capacity and changes in service hours. To keep all staff and physicians informed, the COVID-19 subsite on the Health Centre's intranet was instrumental in linking staff to up-to-date and relevant information regarding the evolution of the pandemic.

Several teams also gathered department-specific data to inform their decision-making and information dissemination. These teams included the Airway Management Group (intubation for COVID-19 patients), Mental Health and Addictions (service changes and usage), human resources (changes in staffing), and Strategy & Organizational Performance team (weekly PPE reports). Additionally, efforts were made to link data provincially, with the Health Authority's'

System Performance and Analytics Teams collaborating to develop the COVID-19 Dashboards.

### LHS characteristic 3: Timely production of research evidence

In response to the pandemic, the health centre participated in a provincial funding initiative to support efforts to generate evidence to address a range of research questions relevant to COVID-19. Seven COVID-19-related studies were launched as part of the province's COVID-19 Health Research Coalition in the areas of Discovery Science, Health System Improvements and Social Sciences. The health centre's Research Services Office collaborated with other pediatric and women's centres across Canada to develop a protocol to quickly close non-COVID related research, employing a work from home strategy for health service researchers and modifying the Research Ethics Board approval process to expedite COVID-19 related studies. While the studies funded through the provincial initiative addressed key issues related to COVID-19, our findings identified limited formal linkages between and within the healthcare community and research community for timely sharing of research evidence to support policy and practice change. Informal communication with trusted sources was identified as the most prevalent strategy for knowledge exchange during early stages of the pandemic.

### LHS characteristic 4: Appropriate decision supports

The Health Centre relied on new and existing *decision-support systems* in their pandemic response. Provincially, the Health Centre is a member of key working groups set up by the Department of Health and Wellness through Public Health with the office of the Medical Officer of Health which guided provincial health system readiness. Locally, PPE tracking systems and work from home guidelines were developed to guide decision-making in these areas. Although the existing, pre-COVID-19 Pandemic Response plan provided some logistical information related to system response, it was not used to guide the specific organization-level COVID-19 response as it contained high level suggestions which did not cover the full breadth of the required response.

### LHS characteristic 5: Aligned governance, financial and delivery arrangements

Throughout the pandemic response, *systems shifted to align* with national, provincial and local decisions and directives. To ensure success of these changes, teams adapted directives to meet the specific needs of the organization and its patient population (i.e., children, women, and youth). The People and Technology committee worked with unions to facilitate rapid staffing changes and redeployment brought about by the pandemic response and supported staff who shifted to working from home. To support financial arrangements and delivery, business continuity planning was initiated for all departments in order to further adjust to the rapid changes brought about by the pandemic.

### LHS characteristic 6: Culture of rapid learning and improvement

The COVID-19 pandemic created a *culture of rapid learning and improvement* in order to respond to the fast-paced changes needed to curb the spread of the virus (i.e., physical distancing measures, increased use of PPE, visitation restrictions). The Health Centre worked closely with provincial organizations and governing bodies to share pandemic-related evidence, develop actions and implement key decisions. Rapid changes were made to the delivery of virtual care, with Mental Health and Addictions Services being recognized as a leader in this area.

Staff and patient feedback were brought to the IMC, facilitating open discussion and helping to maintain an awareness of patient needs among staff.

### LHS characteristic 7: Competencies for rapid learning and improvement

Fear and uncertainty related to the COVID-19 virus, including an anticipated surge in hospitalizations and transmission of the virus amongst patients and providers, facilitated organizational *capacity for rapid learning and improvement*. The pandemic response created a unified objective for the Health Centre which was enacted by all staff at all levels of the organization. To address unprecedented challenges, the Leadership Team coordinated the pandemic response by: a) collaborating with provincial organizations and governing bodies; b) creating new committees (i.e., COVID-19 response committee); and c) leveraging existing teams (People & Organization Development, Logistics and Resources Committee, Clinical Program Operation Committees, etc.). Looking ahead, the Leadership Team developed the *Reimagining and Resuming Services Plan*, which is a commitment to shift operations back to pre-pandemic functioning while remaining agile to re-implement COVID-19 restrictions across the organization during subsequent waves of the pandemic.

## Discussion

This study used a LHS framework to identify assets and gaps in the COVID-19 pandemic planning and response work at a Canadian women and children's tertiary health centre during the initial stages of the pandemic (up to August 31st, 2020). A LHS includes cycles of continuous learning and offers a valuable framework to organize a systematic and data-driven response to health system crises like COVID-19 [7]. Our study examined data from multiple sources and identified several opportunities to improve the LHS infrastructure. This section to follow describes key findings related to the LHS dimensions, as well as practice and research implications related to LHS.

### Engaging patients in rapid decision-making

LHS are anchored on patient needs, perspectives and aspirations [9]. Engaging patients in health research and health care delivery has seen exponential growth in recent years [26]. Aligning communication strategies with the principles of patient engagement and patient- and family-centered care has been identified as critically important during the COVID-19 pandemic [27]. The Health Centre in this study had well-established structures and mechanisms for engaging patients and families such as a Family Leadership Council, a Youth Advisory Council, as well as an established practice of including parent and youth in research. Engaging patients and families in co-creating care is also outlined in the health centre mission statement. However, due to uncertainty related to scarce and evolving evidence related to COVID-19 and the rapid pace of decision-making required to managed the pandemic, many of the usual ways of working based upon patient and family-centred care principles were limited during the first phase of pandemic planning and response [28, 29]. As in many health care organizations, non-essential services and personnel were moved to work-from-home or furloughed. In our study, communicating changes to patients and families regarding how to access care was a key priority for the Health Centre. However, balancing communication of general access policies with tailored messages for special circumstances proved challenging. Patients and families need to be involved in designing care in complex situations such as a pandemic response to ensure care is patient centered [30, p. 7]. The visitor restrictions and physical distancing measures that were implemented proved challenging for some parents and patients who felt isolated from their support network and struggled to build trusting relationships with their care

providers. This can have significant impact on patient and health outcomes; the inability to see, touch and talk to loved ones during a hospital stay can increase the burden of illness [31].

However, as Hart et al. [32] recommend, restrictions on family presence does not need to replace the principles of family-centred care. Moving forward, public and patient engagement will be critical for decision-making about removing COVID-19 restrictions [33]. Similar to how workplace communications have shifted drastically to online communications, patients and families can be engaged via teleconference and videoconference methods in both planning and care delivery. These strategies are needed to support continued pandemic response, as well as planning for post-pandemic health care delivery [34]. Engaging patients and families in this way will address the ethical imperatives and economic and social benefits from patient engagement [35, 36] and strengthen a LHS structure for future rapid-learning and health system change. For subsequent waves of the pandemic and as we move forward post-pandemic, efforts are needed to format feedback channels to better facilitate management and leadership response to pertinent issues and develop a mechanism to support tailored communication to patients and families.

## Improved digital capture, linkage and timely sharing of relevant data

A key component of a LHS is digital capture, linkage, and timely sharing of data (patient experiences, provider outcomes, and other process and outcome indicators), to make timely, evidence-informed decisions [9]. In this study, administrators and health care providers worked quickly to capture, link and share local contextual data related to COVID-19. Several working groups and new teams were organized. However, there was limited interdepartmental sharing of these data and integration of patient experience data into decision-making. There was a stronger focus on broader-level systems data (i.e., PPE use, volume of patients in pandemic assessment centre, human resources re-deployment etc.). A lack of an existing data capture system and the pace of new knowledge led to more reactive initiatives in response to the pandemic and lack of capacity for sharing data, whereas having a comprehensive decision support system, including an electronic health record (EHR), could have supported a proactive response to the pandemic.

Previous research demonstrates the ability of EHRs to capture, link and share data. EHRs with decision support system capabilities have shown to improve patient safety, preventative care, implementation of evidence-based care guidelines, and communication and management of clinical information for providers and patients [37]. EHRs allow for predictive models to be embedded within clinical decision supports to allow for real-time risk prediction and support decision-making [38]. In addition to EHR and decision support systems, a LHS will not be realized without adequate digital capture of the care experience. This includes infrastructure that allows for collection and integration of patient reported experience measures and patient reported outcome measures [39].

Our findings suggest that during early stages of the pandemic, limited real-time health outcomes and experience data were collected to inform rapid decision-making. Further, limitations with provincial information technology support systems meant that significant manual work from decision support services was required during the first wave to generate reports to guide decision-making. In a priority setting exercise to inform Canada's response to the COVID-19 pandemic, McMahon et al. [40] identified the need for timely access to data for researchers, decision-makers, and front-line care providers to inform policy and care delivery decisions, including the rapid analysis of effective and evidence-informed response strategies. COVID-19 has highlighted critical gaps in data capture across Canada, including a lack of ability to link data and collection of race and ethnicity data, which risks further impacts of

pandemic policies on existing health and social inequities [40]. Efforts are urgently needed to build a digital infrastructure that includes care experience data, process and outcome indicators, to inform rapid cycles of policy and care delivery decisions. Inequities related to digital literacy and digital poverty must be considered alongside the development, implementation and evaluation of digital infrastructure changes in the health system.

## Role of embedded research for timely production of evidence

Our study identified a gap in the Health Centre's ability to rapidly generate and incorporate research evidence to support policy and practice decisions related to COVID-19. The Research Services Office quickly focused on the critical administrative tasks of halting non-COVID-19 related research studies and streamlining Research Ethics Board processes to rapidly support projects related to the treatment of COVID-19. While the early focus of research production was on the treatment of COVID-19, members of the Executive Leadership also recognized the impact that the pandemic measures could have on patients, families, health centre staff and providers. Consequently, they collaborated with a provincial funding initiative to commission work to study the impact.

Several factors contribute to the gap in generating and incorporating research evidence into policy and practice decisions. First, this was an unprecedented event with limited published research evidence available to guide policy and practice change, particularly in the early phases of the pandemic. Second, the existing health system-research structures and partnerships that support the timely inclusion of evidence into decision-making were not well established. To be most effective in supporting a LHS, *"researchers must be fully integrated into their internal environments where health problems are articulated, priorities and plans set, new initiatives developed and launched, and resultant changes managed"* [10]. Translation of research into practice can be challenging but having researchers and research programs embedded in health system operations promote direct implementation of evidence-based practices [41]. Moving forward, there is a need to build and strengthen partnerships with health service researchers and implementation scientists internal and external to the health centre to allow for ready access to best available evidence and support the design and evaluation of policy and practice change strategies. Implementation researchers working in collaboration with health system partners can rapidly scale up and spread promising practices to address the changing needs of patients, health care providers, and the health system. To actualize a LHS moving forward, there is an opportunity for novel integrated systems where embedded researchers inform decision-making processes through timely production of evidence.

## Ethical framework for learning health systems

Participants revealed tensions as patients, families, and health care providers experienced the impact of policies and practices deployed throughout the first wave of the pandemic. For instance, health care provider participants identified the ethical and moral dilemmas that were experienced when enforcing visitor restrictions to prevent transmission of the virus. Other research has identified the need to examine the ethical implications of restrictive public health and physical distancing measures, use of technology and data for contact tracing, and the impact of guidelines on equity-seeking populations [40]. Ethical considerations are not included as a main characteristic of Lavis et al.'s LHS framework [9]. Comparatively, Menear et al. [42] developed a framework for value-creating LHS in which an ethical component is described as a main LHS pillar. Given the ethical implications of many COVID-19 responses, and ethical component seems like a timely addition to LHS frameworks to support challenging decision-making.

### Use of LHS as a framework to study implementation

A LHS framework provides an opportunity to enhance health systems, such as the participating Health Centre, to achieve optimal patient outcomes [9]. While LHS are a relatively novel approach to health care, early evidence indicates its effectiveness in supporting health care providers to reduce diagnostic errors [43] and improve patient safety by enhancing interprofessional collaboration to reduce medication errors [44]. Overall, the literature primarily focuses on LHS theory rather than its applicability in practice [39]. To address this limitation, Lavis et al. [9] utilized a LHS framework to map assets and gaps in provincial health systems across their ability to meet the care needs of patients, providers, etc. Similarly, Polancich et al. [45] used a LHS framework to examine the impact of the COVID-19 pandemic on the incidence of hospital-acquired pressure injuries.

Building on Lavis' approach, we used their LHS characteristics as a framework for mapping the assets and gaps, through quantitative and qualitative data sources, in the Health Centre's response to the COVID-19 pandemic. Our evidence suggests that the organization was already implementing many features of a LHS pre-pandemic and has the capacity and infrastructure to further develop as a LHS without radically altering the way it functions (i.e., leveraging existing assets). Moreover, the COVID-19 pandemic has helped accelerate the Health Centre as a functioning LHS. Our study provides an example of applying a LHS lens to analyzing health system decision-making and identifying key components needed to achieve desired patient and health system outcomes. To move the science forward on LHS, efforts are needed to build on existing theories and schematic frameworks and provide practical guidance to researchers and health system decision-makers on how to actualize a LHS in practice. More specifically, research is needed to develop measurement tools, implementation strategies for LHS adoption, LHS indicators in practice and policy, and evaluation measures to understand the impact of a LHS on patient and health system outcomes.

### Strengths and limitations

A key strength of this study was the use of a pre-defined triangulation protocol that supported a systematic method to data mapping. Further, we employed an integrated knowledge translation approach which helped to incorporate multiple perspectives into the data integration process and important insights into health system assets and gaps. Despite these strengths, this study should be considered with the following limitations in mind. First, this study was conducted in the early stages of the pandemic (up to August 31st, 2020); as there has been significant change since that time, our findings may not be applicable to more recent COVID-19 health system experiences. However, the LHS analysis provides high level strategic direction for health systems moving forward, regardless of stage of the pandemic. Second, we focused on one health system; as such, these findings may not be applicable to other contexts. However, this health system provides care to a range of patient populations across a diverse geographic setting. We included rich description of the triangulated data to support its potential application in other settings. Third, the study was descriptive in nature; as such, we are unable to assess how the pandemic preparedness and planning actions worked or did not work. This study could be further strengthened with the inclusion of a comparator health system to evaluate similarities and differences within the LHS dimensions across organizations with similar constraints. This further reiterates the need for evaluative research to move the field of LHS science forward.

### Conclusion

The COVID-19 pandemic has highlighted the urgent need to develop a LHS informed data-driven response to a public-health crisis and complex health system challenges. This study

used a LHS framework to examine the COVID-19 pandemic planning and preparedness work conducted at a Canadian women's and children's health centre. We identified key assets and gaps related to engaging patients in decision-making, improving digital capture, linkage and sharing of relevant data, and timely production of evidence. Overall, this study identified promising strategies for future pandemic planning and preparedness work. Further, we outlined opportunities to strengthen the LHS infrastructure to promote the rapid integration of evidence and lessons learned from patient experiences into decision-making.

## Supporting information

**S1 File. LHS framework matrix.**
(DOCX)

## Acknowledgments

We would like to acknowledge the participants involved in this research, including patients, caregivers, health care professionals, health centre staff and managers. Without their contribution, this work would not have been possible. We would also like to thank the team members who contributed to the broader program of research on this topic: Karen Turner, LeeAnn Larocque, Laura Cole, Sheri Price, Audrey Steenback, Jeanna Parsons Leigh, Tara Sampalli, Jane Palmer, Jeannette Comeau, Shannon MacPhee and Darlene Inglis.

## Author Contributions

**Conceptualization:** Christine Cassidy, Douglas Sinclair, Annette Elliott Rose, Shauna Best, Janet A. Curran.

**Formal analysis:** Christine Cassidy, Meaghan Sim, Mari Somerville, Daniel Crowther, Janet A. Curran.

**Investigation:** Christine Cassidy, Meaghan Sim, Daniel Crowther, Janet A. Curran.

**Supervision:** Christine Cassidy, Meaghan Sim, Douglas Sinclair, Annette Elliott Rose, Stacy Burgess, Shauna Best, Janet A. Curran.

**Visualization:** Mari Somerville.

**Writing – original draft:** Christine Cassidy, Meaghan Sim, Mari Somerville, Janet A. Curran.

**Writing – review & editing:** Christine Cassidy, Meaghan Sim, Mari Somerville, Daniel Crowther, Douglas Sinclair, Annette Elliott Rose, Stacy Burgess, Shauna Best, Janet A. Curran.

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
