## [Decision Letter · Decision Letter 0]

7 Jan 2022

PONE-D-21-15367Using a Learning Health System Framework to Examine COVID-19 Pandemic Planning and ResponsePLOS ONE

Dear Dr. Curran,

Thank you for submitting your manuscript to PLOS ONE. After careful consideration, we feel that it has merit but does not fully meet PLOS ONE’s publication criteria as it currently stands. Therefore, we invite you to submit a revised version of the manuscript that addresses the points raised during the review process.

Two reviewers have provided comments. Both note that additional details need to be added to enable this to be a stand alone publication.

We look forward to receiving your revised manuscript.

Kind regards,

Nancy Beam, PhD

Staff Editor

PLOS ONE

Journal Requirements:

2. Thank you for including your ethics statement: 'REB approval (Institutional approval #1025812) was obtained prior to commencement of data collection.'

3. Please provide additional details regarding participant consent. In the ethics statement in the Methods and online submission information, please ensure that you have specified (i) whether consent was informed and (ii) what type you obtained (for instance, written or verbal, and if verbal, how it was documented and witnessed). If your study included minors, state whether you obtained consent from parents or guardians. If the need for consent was waived by the ethics committee, please include this information.

4. Please include a copy of the interview guide used in the study, in both the original language and English, as Supporting Information, or include a citation if it has been published previously.

6. PLOS requires an ORCID iD for the corresponding author in Editorial Manager on papers submitted after December 6th, 2016. Please ensure that you have an ORCID iD and that it is validated in Editorial Manager. To do this, go to ‘Update my Information’ (in the upper left-hand corner of the main menu), and click on the Fetch/Validate link next to the ORCID field. This will take you to the ORCID site and allow you to create a new iD or authenticate a pre-existing iD in Editorial Manager. Please see the following video for instructions on linking an ORCID iD to your Editorial Manager account: https://www.youtube.com/watch?v=_xcclfuvtxQ.

Reviewers' comments:

Reviewer's Responses to Questions

**Comments to the Author**

1. Is the manuscript technically sound, and do the data support the conclusions?

Reviewer #1: Yes

Reviewer #2: Partly

2. Has the statistical analysis been performed appropriately and rigorously? 

Reviewer #1: I Don't Know

Reviewer #2: N/A

3. Have the authors made all data underlying the findings in their manuscript fully available?

Reviewer #1: Yes

Reviewer #2: No

4. Is the manuscript presented in an intelligible fashion and written in standard English?

Reviewer #1: Yes

Reviewer #2: Yes

5. Review Comments to the Author

Reviewer #1: Thank you for allowing me the opportunity to review this manuscript. This manuscript examines the use of a learning health system framework to explore COVID-19 planning and response. This work is both timely and critical to improve future pandemic planning.

ABSTRACT

In the background, I would be cautious to use terminology such as “wave one”, as I am not sure epidemiologists have yet been able to come to a consensus regarding how this pandemic has presented regarding case load.

INTRODUCTION

Line 89: Again, I would be cautious using wave one. Although you have operationalized this as prior to August 2020, I feel this may introduce unneeded controversy. I would instead state something similar to “early response to the COVID-19 pandemic” and include “(up to August 2020”, unless you are able to cite a respected work to back this timeline.

In line 91, the authors state the goal of the paper. Lines 96 through 99 states the aims of the paper as three research questions. This feels a little disjointed. I would instead suggest stating something similar to; “the goal was/is achieved by addressing the following aims/research questions…” to be clear on what is expected in the results of the manuscript.

METHODS

Study Design

Overall, this section should be expanded to provide more transparency in how the study was conducted. This could be improved in the following ways:

1. State why a mixed methods approach was the best way to answer the research questions.

2. As there are several approaches to mixed methods designs (this one choosing Creswell), list what data is being collected and used (administrative and textual data and interviews), when, and how. There is no need to go into depth about the data sources as this is detailed later in the methods, but an overview/preview would be helpful for flow.

3. The authors may want to consider adding a figure. It is stated on line 105 that iterative cycles of data collection/analysis were preformed, but it is not clear when or how. A flow diagram or something similar may be helpful for the reader to visualize the process. In such a figure, cycles of iteration and timing of data collection/analysis may be better understood.

4. The learning health system has an apparent large influence over this study, yet there are no mentions of a LHS framework nor the parallels between the components assessed in this study and the LHS. To be clearer, the authors should expand more on the rigor/trustworthiness of their work in assuring it ties to an LHS “lens”. This may be done through a conceptual model or further explanation.

Study Setting

Please confirm if this is indeed a “single” center, or if there are multiple locations in the system. Would also be interesting to know the approximate patient capacity, if available.

Data Sources

Administrative and textual sources

This explanation of the administrative and textual sources remains unclear until the explanation under the “Data analysis” section. At a minimum, a statement should be made regarding the purpose of choosing this particular data source. To mitigate this confusion, the authors may elect to combine the “Data sources” and “Data analysis” sections into one. Regardless, a clearer understanding of how this data will be used to reach the goals of the research (which also may be more clearly outlined in an overall figure as suggested above) is needed.

As a minor comment, the explanation regarding single reviewer examination and data organization may be better suited in the data analysis section/explanation.

Qualitative Interviews

Line 132: Mention of wave 1 (if changes are made to this verbiage)

Although details may be found in other manuscripts, it is still important to provide some context into the other studies to provide clarity for this analysis. It is also somewhat unclear what was analyzed previously and what (if any new data were generated in a qualitative analysis for this study) was done in the current manuscript regarding the interviews/thematic analysis. Please make this clearer to allow for greater transparency.

I would suggest that the interview guide be included as a supplementary material/appendix.

DATA ANALYSIS

Administrative and textual data sources

Between this section and “Data triangulation”, I don’t feel that there is adequate explanation of how the grouping the documents into three categories helped to facilitate a mixed-methods analysis. Although later the authors describe the development of a matrix structure, the purpose of examining the documents is unclear. What, if anything, was extracted from these documents? If they were simply grouped according to pillars of infection prevention and control/federal COVID guidelines, why not simply use these as guidance during the qualitative analysis (especially if no new categories emerged)?

Qualitative interviews

Line 153 should be deleted.

Although a method (directed content analysis) is cited, further detail of this method should be included. This could be a numbered list or descriptive text, but should address the general steps of the analysis, how many coders there were, if inter-rater reliability was calculated/assessed, assurances of trustworthiness, etc.

Data triangulation

Line 167-168- “through a series of individual and group tasks”. Such as?

RESULTS

Line 183- Again, even though the results are published elsewhere, there still should be some information available in this manuscript to make it a standalone paper. Qualitative themes could potentially be referenced here or included as a supplementary material/appendix. The number of participants/participant roles would also be helpful.

The following information in the results and the discussion are well written and interesting. However, I am having a difficult time drawing parallels between Figure 2, the subheadings in the results, and the subheadings in the discussion. It appears as there are three different interpretations to the data. The authors should make some effort to reorganize/relabel these sections to help readers with interpretation. Disclosing the matrix in a figure might be helpful. Regardless, the results should succinctly report how the authors came to their LHS-based improvements for future pandemic response, and then elaborate on these ideas and make specific suggestions in the discussion. Based on the three concurrent manifestations of these improvements/findings, I cannot give specific instruction on the best approach.

LIMITATIONS

The authors do not appear to disclose any limitations relevant to their work. This should be completed. One major limitation I have noted is the use of a women’s/children’s hospital as the sole location for data collection. This sample may limit the applicability to other health settings.

TYPOGRAPHICAL AND GRAMMATICAL REVISIONS

Line 61: “for the need to develop of sustainable”- I believe “of” should be omitted.

Line 62: Times should be time

Line 76: Lavis et al. define- should be defines or defined

Line 108: REB should be spelled out

OVERALL COMMENTS

Overall, this manuscript reads well and is in an area of interest to the journal’s audience. The findings are interesting but would benefit from greater clarity in the methods and a reorganization of the findings and discussion. Also please make sure to disclose relevant information from the other two studies your team has conducted- although some information may be available in other manuscripts, having such information available in this manuscript is essential so that it may be viewed as a stand-alone study.

Again, thank you to the authors for conducting this timely and relevant work, and to the editorial team at PLOS ONE for inviting me to review this manuscript.

Reviewer #2: See attachment.

The documents refers to data that are not available to the reader to date, which makes the results difficult to appreciate. I would like to suggest to include at least some summarized data about these other papers.

What quality standards were followed to ensure the quality of the mixed method described in the document?

6. PLOS authors have the option to publish the peer review history of their article (what does this mean?). If published, this will include your full peer review and any attached files.

Reviewer #1: No

Reviewer #2: **Yes: **Thérèse Van Durme

---

## [Author Response · Author response to Decision Letter 0]

7 Apr 2022

Please see attached cover letter with detailed response to reviewers' comments.

---

## [Decision Letter · Decision Letter 1]

30 May 2022

PONE-D-21-15367R1

Using a Learning Health System Framework to Examine COVID-19 Pandemic Planning and Response

PLOS ONE

Dear Dr. Curran,

Thank you for submitting your manuscript to PLOS ONE. After careful consideration, we feel that it has merit but does not fully meet PLOS ONE’s publication criteria as it currently stands. Therefore, we invite you to submit a revised version of the manuscript that addresses the points raised during the review process.

Please see the comments below for both the reviewers comments and my suggestions for revision.

We look forward to receiving your revised manuscript.

Kind regards,

Brian J Douthit, PhD, RN-BC

Guest Editor

PLOS ONE

Journal Requirements:

Additional Editor Comments (if provided):

Thank you for your patience through this review process. For full transparency, I had previously served as a reviewer for this paper and been asked by the journal to serve as guest academic editor.

Based on the comments of the reviewers, I feel as though there could be some improvement to the paper without the need to rework your methodology. In short, please review the comments carefully and edit the manuscript to address the concerns of the reviewers as thoroughly as possible. Specifically, please be sure to be as transparent in your methods as possible (as one reviewer notes, you should list the number of participants in the interviews, etc.). I would also argue that this is not 'mixed-methods', but 'multi-methods'. Mixed-methods research is a specific methodology which may have thrown some of the reviewers off. Also please be sure to update the limitations, using some of the critiques as a guide. There is also some good advice on how to better frame your discussion as outlined by reviewer 3.

Thank you again for your submission, and I am looking forward to your revision.

Reviewers' comments:

Reviewer's Responses to Questions

**Comments to the Author**

1. If the authors have adequately addressed your comments raised in a previous round of review and you feel that this manuscript is now acceptable for publication, you may indicate that here to bypass the “Comments to the Author” section, enter your conflict of interest statement in the “Confidential to Editor” section, and submit your "Accept" recommendation.

Reviewer #3: (No Response)

Reviewer #4: All comments have been addressed

2. Is the manuscript technically sound, and do the data support the conclusions?

Reviewer #3: No

Reviewer #4: Yes

3. Has the statistical analysis been performed appropriately and rigorously? 

Reviewer #3: N/A

Reviewer #4: N/A

4. Have the authors made all data underlying the findings in their manuscript fully available?

Reviewer #3: (No Response)

Reviewer #4: Yes

5. Is the manuscript presented in an intelligible fashion and written in standard English?

Reviewer #3: Yes

Reviewer #4: Yes

6. Review Comments to the Author

Reviewer #3: The paper concerns the use of a Learning Health System framework to study the covid-19 response in a Canadian health centre. I have a number of concerns:

* the paper is extremely vague on the methodology. It claims to use a mixed methods approach but it is very hard to get a sense of the nature or volume of data or sampling logic on the either the qual or quant part (how many interviews? what was this "health administrative and human resource data"?)

* the findings do not cross reference back to the data, and so it is difficult to judge the strength of the evidence based underlying the narrative. the findings often relate to things which were done (eg there was a covid subsite for sharing information) rather than rounded assessments of whether they worked. Many of the findings statements are very vague (eg "through the pandemic, systems shifted to align with national, provincial and local decisions and directives"; "the pandemic response created a unified objective for the health centre which was enacted by all staff at levels of the organization") and sound more like the self-congratulatory stories which managers tell themselves rather than true evidence-based research findings.

* The discussion does not clearly align with the findings. Partly this is a problem of structure: Given that the LHS concept has 7 dimensions, I would expect that the discussion would be organised along similar lines, but some of the subheadings relate to some dimensions and others do not. But it is also a problem of content: for example the section on EHRs gives reasons why EHRs are a good thing but these are reasons which could have been given without doing this study.

* A major weakness of the study design is that lack of a comparator. Ultimately all organisations are constrained by resources and structural factors. It's very hard to assess whether the current study setting did well or badly on the LHS dimensions without reference to what was done by other comparable entities which shared the same constraints.

Reviewer #4: In this manuscript, the authors provide detailed information on the application of a LHS framework in understanding pandemic planning and response during the early stages of the COVID-19 pandemic in a single-site in Canada. A previous draft of the manuscript has undergone review and the authors have made major revisions, successfully addressing the comments provided by the reviewers. This has resulted in a clearer and more focused manuscript. I have noted some suggestions for further minor revisions below:

1. While there is valuable learning from this study, particularly for other OECD countries, it is important to note that there is wide variation in the organisation, structure, and culture of health care systems and the application of the LHS approach may not be feasible or useful everywhere.

a. It should be clear earlier on in the manuscript that this is a single-site study at a Canadian hospital (e.g. in the title and abstract).

b. It would also be useful to provide some additional information for an international audience relating to the context of this healthcare system, e.g. insurance vs. publicly funded, rural/urban population served, key population-level demographic data for the area served such as socio-economic deprivation, age distribution, ethnicity.

2. P. 5, line 102, fortnightly may be a better term than bi-weekly.

3. P. 5, line 104, there is an additional full stop.

4. P. 16, digital communication has been central to continuing provision of healthcare services during the pandemic, but consideration needs to be given to the potential impact of this on patient-centred care and to health inequalities, notably with regard to digital literacy and digital poverty.

7. PLOS authors have the option to publish the peer review history of their article (what does this mean?). If published, this will include your full peer review and any attached files.

Reviewer #3: No

Reviewer #4: No

---

## [Author Response · Author response to Decision Letter 1]

2 Aug 2022

Please see attached document for response to reviewers' comments.

---

## [Editor Report · Decision Letter 2]

4 Aug 2022

Using a Learning Health System Framework to Examine COVID-19 Pandemic Planning and Response at a Canadian Health Centre

PONE-D-21-15367R2

Dear Dr. Curran,

We’re pleased to inform you that your manuscript has been judged scientifically suitable for publication and will be formally accepted for publication once it meets all outstanding technical requirements.

Kind regards,

Brian J Douthit, PhD, RN-BC

Guest Editor

PLOS ONE

Additional Editor Comments (optional):

Thank you for your thoughtful responses to the reviewers. The manuscript reads much more smoothly and presents the findings in a much more organized. Thank you again for your patience and willingness to resubmit.

---

## [Editor Report · Acceptance letter]

30 Aug 2022

PONE-D-21-15367R2 

Using a Learning Health System Framework to Examine COVID-19 Pandemic Planning and Response at a Canadian Health Centre 

Dear Dr. Curran:

I'm pleased to inform you that your manuscript has been deemed suitable for publication in PLOS ONE. Congratulations! Your manuscript is now with our production department. 

Kind regards, 

on behalf of

Dr. Brian J Douthit 

Guest Editor

PLOS ONE